# What Else Do I Need to Know?
# The Effect of Background Information on Users' Reliance on QA Systems

**Navita Goyal**[1], **Eleftheria Briakou**[2][*] **Amanda Liu**[1], **Connor Baumler**[1],
**Claire Bonial**[3], **Jeffrey Micher**[3], **Clare R. Voss**[3], **Marine Carpuat**[1], **Hal Daumé III**[1,4]

[1]University of Maryland, [2]Google, [3]U.S. Army Research Lab, [4]Microsoft Research
navita@cs.umd.edu, ebriakou@google.com, amandastephanieliu@gmail.com,
baumler@cs.umd.edu, {claire.n.bonial, jeffrey.c.micher}.civ@army.mil,
clare.r.voss.civ@army.mil, marine@umd.edu, me@hal3.name

## Abstract

NLP systems have shown impressive performance at answering questions by retrieving relevant context. However, with the increasingly large models, it is impossible and often undesirable to constrain models' knowledge or reasoning to only the retrieved context. This leads to a mismatch between the information that *the models* access to derive the answer and the information that is available to *the user* to assess the model predicted answer. In this work, we study how users interact with QA systems in the absence of sufficient information to assess their predictions. Further, we ask whether adding the requisite background helps mitigate users' over-reliance on predictions. Our study reveals that users rely on model predictions even in the absence of sufficient information needed to assess the model's correctness. Providing the relevant background, however, helps users better catch model errors, reducing over-reliance on incorrect predictions. On the flip side, background information also increases users' confidence in their accurate as well as inaccurate judgments. Our work highlights that supporting users' verification of QA predictions is an important, yet challenging, problem.

## 1 Introduction

With the advent of large language models pretrained on massive data, question answering (QA) systems are able to reason about information that is external to their input context, based on their implicit factual or commonsense knowledge (Petroni et al., 2019; Jiang et al., 2020) or by employing shortcuts to get to the correct answer (Min et al., 2019; Chen and Durrett, 2019; Trivedi et al., 2020). However, in such incomplete information settings, users may lack important background knowledge required to assess the correctness of model predictions: consider the example of Figure 1. Here, a

user interacts with a QA model to seek information about which non-Swedish actress starred in the movie "Light Between Oceans." The QA model retrieves the context *(left)* about the movie and predicts *Alicia Vikander* as an answer based on this context. In such workflows, the retrieved context serves as supporting evidence for the predicted answer. However, in order to assess the accuracy of this prediction, the user must already know that Vikander is Swedish, a piece of information that is not present in the provided context. This leads to a knowledge gap between the information that is required to answer the question and the information that is made available to the user.

Chat-based interfaces are becoming increasingly popular for information-seeking. In the case of factoid questions, these systems commonly surface retrieved or generated information that supports the provided answer. This supporting information is commonly seen through the lens of "information necessary for the underlying model to make the correct prediction." However, this does not necessarily translate to "information sufficient for users to assess the prediction." Recent work in explainable NLP (Fok and Weld, 2023; Xie et al., 2022) argues that to support users in decision-making and foster appropriate reliance, we should focus on providing users with information that aids them in assessing the correctness of model predictions. However, the effects of including such additional information on users' reliance on a QA model are not well studied.

In this work, we design a Wizard-of-Oz experiment to study how users interact with a QA model when some of the information required to assess the correctness of the model prediction is missing. Further, we investigate how *users' reliance on model predictions* and *confidence in their judgments* change when the requisite background—the information required to assess the correctness of the prediction—is provided to the user. The background serves as an extension of the model's in-

---

[*]Work done while at the University of Maryland.

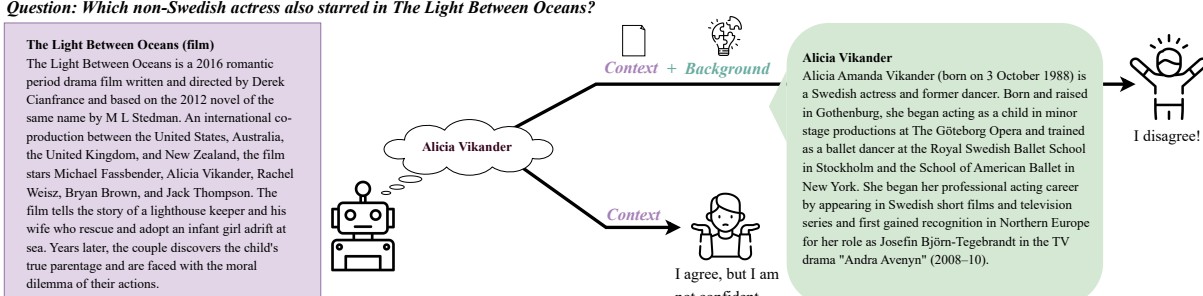

Figure 1: A QA model may make a prediction based on insufficient context (left), making it difficult for users to assess its correctness. Providing the necessary **background** information (right) might reduce the user's over-reliance.

put context and prediction where the goal is not to *justify* the prediction but rather to provide additional context that may establish a common ground between users and the model, a prerequisite for effective collaboration (Bansal et al., 2019).

Further, we study whether the cognitive effort in consuming background mediates the effect of background on users' reliance on model predictions and confidence in their judgments. We employ highlights to surface the important parts of the context and the background to the user, making these easier to consume. We assess whether this highlighting of the relevant content helps users better rely on the model and calibrate their confidence.

We control the sufficiency of the information shown to users by employing a multi-hop question-answering dataset. We consider a QA model that predicts the answer to a complex question based on an input context that alone is insufficient to answer the question without additional background knowledge. Using this model, we design a user-study where we test how the user's agreement with the predicted answers changes when only part or all of the information is disclosed to the users. Our findings are summarized as follows:

○ In line with previous findings on users' over-reliance on AI predictions (Bansal et al., 2021; Bussone et al., 2015; Jacobs et al., 2021; Lai and Tan, 2019), users over-rely on model predictions even in the absence of the relevant background required to assess their correctness.

○ Adding the requisite background helps users better identify incorrect model predictions, significantly reducing over-reliance on the model.

○ Background also significantly increases users' confidence in their own judgments, both when they are correct and when they are incorrect.

○ Including highlights with background does not reduce users' over-reliance on the model or alle-

viate the issue of overconfidence. This indicates the need to further examine the question of how to best aid users in AI-assisted decision-making.

## 2 Research Questions

We conduct an online user-study[1] based on Wizard-of-Oz experiments to examine how users' reliance on model predictions and confidence in their judgments change based on whether they are given the background information required to verify the predictions of a QA model. Our study targets four key research questions:

***RQ1: How do users interact with the model predictions in the absence of sufficient information to assess their correctness?*** Human-AI decision-making often relies on humans to critically assess model predictions instead of simply agreeing with the model (Bussone et al., 2015; Buçinca et al., 2021; Green and Chen, 2019). Users may frequently lack the necessary background to assess the correctness of the model. In such cases, we might expect users to either disagree with the model or have low confidence in their judgments. We study whether users indeed calibrate their confidence and reliance on the model when they likely lack enough information to assess the veracity of its predictions.

***RQ2: Does adding requisite background information allow users to calibrate reliance and confidence on model predictions?*** Building on RQ1, we study whether having access to information required to assess model predictions helps users make more informed decisions with higher confidence. We examine whether users are able to appropriately calibrate their reliance on models when provided with the relevant information to discern correct and incorrect model predictions.

---

[1]The study was approved by the University of Maryland Institutional Review Board (IRB number 1941629-1).

***RQ3: Are users able to calibrate their reliance and confidence even when not all background provided is perfect?*** Extending RQ2, which explores users' reliance and confidence when provided with a background that enables perfect verification of model predictions, we study whether users are able to calibrate their reliance and confidence in model predictions when the background is sometimes, but not always, perfect. This reflects a more realistic scenario where users might be at times provided with background information that is *ungermane*: on topic, but not directly useful for answering the question. Comparing ungermane and germane backgrounds also helps in unconfounding the effect of the presence of more information vs the effect of useful background.

***RQ4: Does highlighting important parts of context and background improve users' reliance and confidence calibration?*** Background information, although potentially useful, also increases the information load and cognitive burden on the end user (Kaur et al., 2020). Thus, the effect of the presence of background is potentially mediated by the effort required in consuming the said background. To assess the mediating effect of the effort required to consume background, we introduce highlights in both the context and the background, identifying sentences that are crucial to answering the question.

## 3 Designing a Wizard-of-Oz System by Using a Multi-Hop Dataset

We design a Wizard-of-Oz study with insufficient (no background) and sufficient (with background) information conditions. We employ a multi-hop question answering dataset, HotpotQA (Yang et al., 2018), which requires multiple pieces of evidence to reason the answer. We repurpose the HotpotQA dataset to control the information available to the model by only surfacing partial information to examine RQ1 and providing the missing information to examine RQ2. To approximate a near-realistic setting, we consider a subset of HotpotQA on which the model predictions remain unchanged when the background is added to the input to the model. This mimics a scenario where a QA model can make a prediction based on partial information using heuristic shortcuts (Min et al., 2019; Chen and Durrett, 2019; Trivedi et al., 2020) or implicit knowledge acquired through pre-training (Petroni et al., 2019; Jiang et al., 2020), but users would lack sufficient background to assess its correctness.

Each question in the HotpotQA dataset is associated with two or more relevant context paragraphs: the context paragraph containing the gold-standard answer to the complex question and the other intermediate paragraphs providing background context corresponding to the required reasoning steps implicit in the question. We use the context paragraph with the gold-standard answer as the supporting context in the *no background condition* (RQ1) and include the background context along with the supporting document in the *with background condition* (RQ2). We only consider questions that involve exactly two documents to ensure that the relevant background is sufficient to assess the model prediction. For the *mixed background condition* (RQ3), we sample a distractor document that is retrieved using the question, but that is not the required supporting document for answering the multi-hop question as the ungermane background. Further, the HotpotQA dataset identifies the sentences in each paragraph that are essential *supporting facts* for answering the question. We use these supporting facts to highlight the context and the background information in the *with background and highlights condition* (RQ4). See Table 1 in the Appendix for examples.

In realistic scenarios, the background information and highlights would be automatically retrieved or generated. However, for the purposes of this study, automating background and highlight extraction confounds the utility of the information presented to the end-user with the performance of the automated system. The proposed Wizard-of-Oz setting allows us more control over the background and highlights exposed to the end-user. This allows us to make stronger conclusions on whether and how users make use of additional information when it is known to be necessary and sufficient to assess model predictions in advance of future research on how to best extract and evaluate such information automatically.

We choose a question answering model with the following desiderata: we want a model that (1) is a realistic representative of a standard QA model and (2) can make correct predictions based on its implicit background knowledge (as acquired from pre-training data), even when given questions and partial context, that is insufficient to answer the questions. Based on this criteria, we select a 336M parameter BERT model (Devlin et al., 2019) fine-tuned on the Stanford Question Answering

Dataset (SQuAD) (Rajpurkar et al., 2016), which consists of extractive question answering pairs over Wikipedia context paragraphs. We employ a single-hop question answering model, as opposed to a model fine-tuned on a multi-hop QA task, as the underlying assumption is that the model is able to answer the complex question, even without the background information. The BERT model fine-tuned on SQuAD achieves an accuracy of $\sim76\%$ on the development set of HotpotQA both in the presence and absence of complete input context that is, in principle, needed to answer the question.

## 3.1 Study design

*Conditions.* We design four conditions by varying the information that is provided to the participants:
- Without background and no highlights ("*No background*");
- With germane background and no highlights ("*With background*");
- With a mix of germane and ungermane background and no highlights ("*With mixed background*");
- With germane background and supporting facts highlighted ("*With background and highlights*").

We conduct a between-subjects study with participants randomly assigned to one of these four conditions. We present each participant with 10 questions in the same condition. We select the questions shown to each participant such that the model prediction is correct on 7 questions and incorrect on the remaining 3. We hold this distribution fixed to ensure that model accuracy on the observed examples is close to the true model accuracy ($\sim76\%$) and to avoid any undesirable effect of the observed accuracy on participants' reliance on the model. In the *mixed (germane and ungermane) background* condition, we provide a germane or ungermane background on each question uniformly at random, both for correct and incorrect predictions. We consider the same pool of questions across conditions and only sample each question once per condition.

For each question, we present the participants with the context paragraph from Wikipedia that contains the gold-standard answer and a model predicted answer corresponding to a span of text in the context paragraph. Depending upon the condition they are assigned to, we additionally show the corresponding (germane or ungermane) background information, with or without highlights.[2]

For each question, participants have to indicate: (1) whether they agree or disagree with the model predicted answer and (2) their level of confidence in their judgment, using a 5-point Likert scale. At the end of the study, we collect aggregate ratings from participants on their perception of the utility of background and highlights, if applicable, their confidence in the model, and their self-confidence in their own responses.[3]

**Procedures** We conduct our user-study online with crowd-workers on Prolific.[4] Human-AI interaction for question answering is fairly ecologically valid as lay users frequently interact with QA systems (search engines, chatbots) for information-seeking purposes. Participants are first presented with details about the study to obtain their consent to participate. They are then shown a tutorial introducing relevant terminology (e.g., background, highlights), along with instructions on how to perform the task. After the tutorial, participants are asked to perform the task for 10 questions, one at a time. After completing the ten questions, participants are asked to complete the end survey to assess their overall perception of the system. Participants are also asked to provide optional free-text feedback or comments on the study. Finally, participants are asked to provide optional demographic information, such as age and gender.

We include two attention-check questions during the study, in each case asking participants to indicate their agreement or confidence level in the previous question after they have clicked away.

**Participants** We recruited 100 participants, with each participant restricted to taking the study only once. Participation was restricted to US participants, fluent in English. Out of the 96 participants who completed the study, we discarded the responses from a single participant who failed both attention checks. The study took a median time of $\sim9$ minutes to complete. Each participant was compensated US\$2.25 (at an average rate of US\$15/hour). $42\%$ of the participants self-identified as women, $54\%$ as men, $2\%$ as non-binary/non-conforming and $0\%$ as any other gender identity. $22\%$ of participants were between the ages of 18-25, $43\%$ between 25-40, $28\%$ between 40-60 and $6\%$ over the age of 60.

---

[2]See Appendix E for the study interface details.

[3]See Appendix B for details on the post-task survey.
[4]http://prolific.co

## 3.2 Measures

We measured several aspects of users' behavior during the study, including their agreement with both correct and incorrect model predictions and their confidence in their own judgments. In what follows, we refer to the predictions made by the model as "predictions" and the decisions by the users to agree or disagree with that prediction as "judgments." Users' judgments are subsequently deemed "accurate" when they agree with correct predictions or disagree with incorrect predictions and "inaccurate" when they agree with incorrect predictions or disagree with correct predictions (Jacovi et al., 2021; Vasconcelos et al., 2023). Using this terminology, we define the following measures:

○ *Appropriate agreement:* Percent agreement with correct predictions.

○ *Inappropriate agreement:* Percent agreement with incorrect predictions (over-reliance).

○ *Users' accuracy:* Percent agreement with correct and disagreement with incorrect predictions.

○ *Users' confidence:* Average confidence (on a scale of 1-5) in accurate or inaccurate judgments.

## 4 Results[5]

***RQ1: How do users interact with the model predictions in the absence of sufficient information to assess their correctness?*** As seen in Figure 2, we find that the users have a significantly higher agreement with correct model predictions (appropriate agreement: $0.89 \pm 0.02$) than incorrect model predictions (inappropriate agreement: $0.61 \pm 0.06$). Users also exhibit a significantly higher confidence in their accurate ($4.08 \pm 0.08$) as compared to their inaccurate judgments ($3.37 \pm 0.17$). This indicates that users are able to calibrate their reliance and confidence, even in the absence of the background. Regardless, users' agreement rate is fairly high (over $80\%$) without background, with over $60\%$ agreement rate for incorrect predictions.

Users' confidence rate is also substantially higher than the neutral score of 3 (around $3.91$) without background, with over $3.37$ confidence even when their judgments are inaccurate. This highlights that users frequently rely on incorrect predictions, even when they have insufficient information to assess the prediction, indicating over-

---

[5]To account for multiple testing, we perform a Benjamini-Hochberg correction (Benjamini and Hochberg, 1995) and report "significance" with a false discovery rate of 0.05 (Hu et al., 2010), yielding a significance threshold $p < 0.01$.

reliance on the model. Even when predictions are correct, high reliance without background is still concerning as it indicates that users overly trust the model despite insufficient information. We note an example of this behavior in a user's feedback at the end of the study: *"Some questions asked for 2 things, like the type of game for 2 games, but the article only has one game info in there. However, I made my decision based on what information I can get from the article, **I think most of the time, AI made the right prediction so I chose "certain" about the AI's decision**."*

***RQ2: Does adding requisite background information allow users to calibrate reliance and confidence on model predictions?*** Comparing the users' rate of appropriate and inappropriate agreement with and without background (Figure 2 (left)), we find that adding background information indeed helps combat over-reliance on incorrect predictions; users exhibit a significantly lower ($p=0.01$) rate of agreement on incorrect predictions in the *with background* condition ($0.47 \pm 0.04$) than the *no background* condition ($0.61 \pm 0.04$). Background information does not affect appropriate reliance; that is, the rate of agreement on correct predictions is the same with and without background ($0.88 \pm 0.02$).

Comparing users' accuracy in *no background* and *with background* conditions, we find that the user accuracy in detecting correct vs. incorrect predictions is marginally higher with background ($0.77 \pm 0.02$) than without ($0.73 \pm 0.02$). This is a natural extension of the rate of appropriate and inappropriate agreement as we observe a close agreement rate for correct predictions but a much lower agreement rate for incorrect predictions in the *with background* condition, which results in an overall higher accuracy. However, this effect is not significant ($p=0.14$), perhaps partly because only $30\%$ of the examples a user sees are incorrect.

In terms of the effect of background on users' confidence in their judgments (Figure 2 (right)), we observe that background information increases users' confidence in both their accurate judgments (from $4.05 \pm 0.06$ to $4.49 \pm 0.04$; $p=0.08$) and inaccurate judgments (from $3.55 \pm 0.12$ to $3.89 \pm 0.10$; $p=0.03$). This reflects that although background helps calibrate reliance on the model, it also leads to overconfidence in inaccurate judgments: users exhibit higher confidence in their inaccurate judgments with the background than without. On the whole, users' confidence in their judgments is fairly

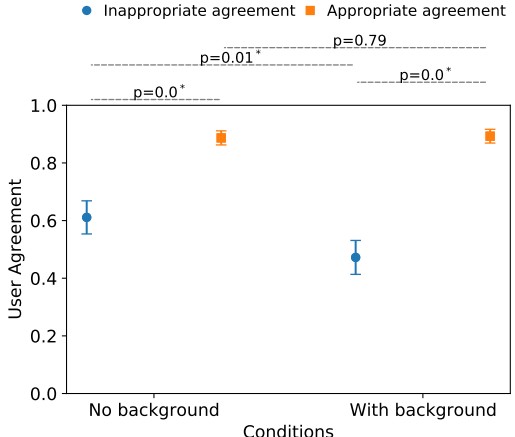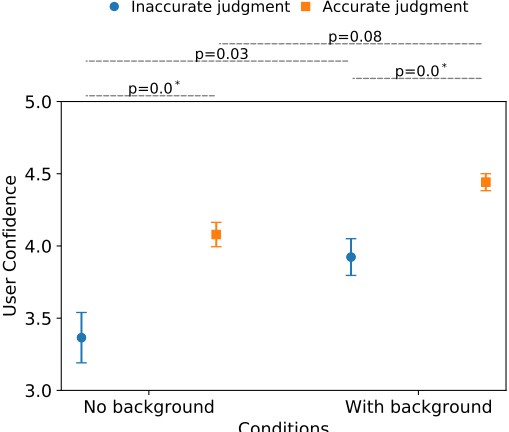

Figure 2: User agreement rate with model predictions when the model is incorrect vs. correct (left) and user confidence in their own judgments when the user judgment is accurate vs. inaccurate (right). The graphs show mean and standard error of agreement/confidence in *with/without background* conditions. The rate of agreement is higher for correct predictions (appropriate agreement) than for incorrect predictions (inappropriate agreement), both with and without background. Users exhibit higher confidence in their accurate judgments than in inaccurate judgments. However, the rate of inappropriate agreement is fairly high (0.6), even without background (*RQ1*). Adding background reduces users' over-reliance on incorrect predictions. However, adding background also increases users' overconfidence in their inaccurate judgments (*RQ2*).[6]

calibrated: users are significantly ($p$=0.0) more confident in their accurate judgments ($4.28 \pm 0.04$) than their inaccurate judgments ($3.71 \pm 0.08$). This is true regardless of background explanations. However, the jump in confidence between inaccurate and accurate judgments is higher with background explanations (Cohen's d: $0.70$) than without (Cohen's d: $0.42$).

***RQ3: Are users able to calibrate their reliance and confidence even when not all background provided is perfect?*** From RQ2, we find that background helps users calibrate their reliance on model predictions. We further test whether users are able to calibrate their reliance and confidence in model predictions even when the background is only useful in a subset of examples shown to the users. Comparing the users' rate of appropriate and inappropriate agreement with germane vs. ungermane background in the *mixed background condition* (Figure 3 (left)), we find that the rate of appropriate agreement is significantly higher ($p$=0.0) when users are shown germane background, which is topical and useful, ($0.93 \pm 0.03$) as compared to when users are shown ungermane background, which is on topic, but not directly useful ($0.74 \pm 0.05$). Further, the rate of inappropriate agreement is also lower with germane background ($0.53 \pm 0.08$) than

with ungermane background ($0.60 \pm 0.08$), however, this difference is not significant ($p$=0.52). This leads to an overall significantly higher ($p$=0.01) accuracy in the examples with germane background ($0.78 \pm 0.04$) than the examples with ungermane background ($0.65 \pm 0.04$).

We also observe that even when the background shown to users is a mix of germane and ungermane backgrounds, users achieve comparable accuracy ($0.77 \pm 0.02$, $p$=0.98) with a germane background as in the *with background condition*. More discussion on these comparisons is included in Appendix C.

In terms of the effect of the relevance of background on users' confidence in their judgments (Figure 3 (right)), we observe that similar to the *no background* condition, users exhibit lower confidence in both their accurate and inaccurate judgments when shown ungermane backgrounds as compared to when shown germane backgrounds.

All together, these results indicate that users indeed pay attention to the information provided in the background, which is reflected in better calibration in model predictions when the background is germane and useful. Despite this, users still exhibit overconfidence in their inaccurate judgments for examples with germane backgrounds, even though the background points out the inaccuracies in the judgments. We conjecture that this is possibly because identifying (and possibly discarding) unger-

---

[6] $*$ indicates significance after Benjamini-Hochberg multiple-testing correction with a false discovery rate of 0.05.

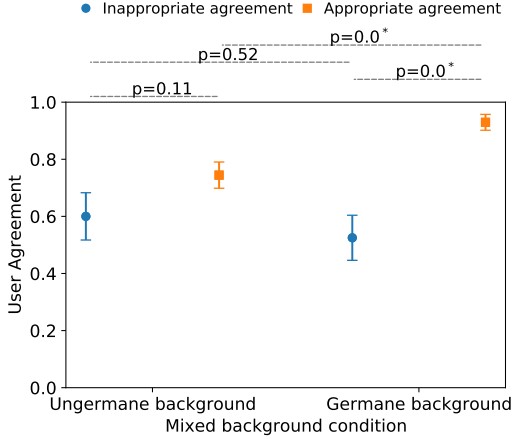
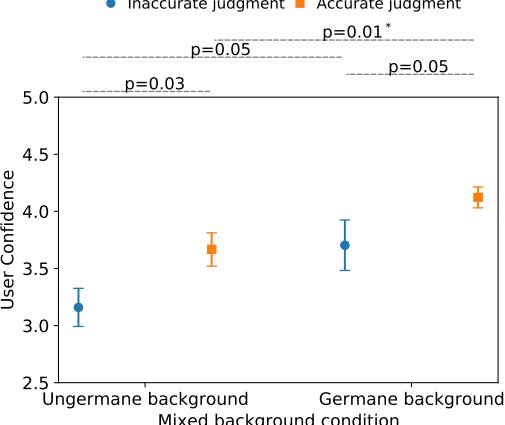

Figure 3: User agreement rate (mean and standard error) with correct vs. incorrect model predictions (left) and user confidence (mean and standard error) in their own accurate vs. inaccurate judgments (right) in the *mixed (germane or ungermane) background* condition on examples with *ungermane background* vs. *germane background*. The rate of appropriate agreement is higher when users are shown a germane background as compared to when they are shown an ungermane background. Users' overconfidence in inaccurate judgments is marginally higher when provided with a germane background as compared to an ungermane background (*RQ3*), similar to no background condition in *RQ2*.[6]

mane background is easier as compared to verifying reasoning errors in model predictions.

***RQ4: Does highlighting important parts of context and background improve users' reliance and confidence calibration?*** From RQ2, we find that background indeed helps in combating over-reliance on model predictions but also leads to over-confidence in the users, even in inaccurate judgments. We test whether highlighting supporting facts helps users better utilize the information present in the background. To this end, we compare the conditions *with background (no highlights)* and *with background and highlights* (Figure 4). We find that adding highlights to the context and background does not reduce users' over-reliance on incorrect model prediction ($0.46 \pm 0.06$) over simply adding the background information ($0.47 \pm 0.04$).

Further, highlighting relevant parts of the context and background does not alleviate the issue of over-confidence stemming from the presence of background information: users' confidence in their incorrect judgments is similarly high in the condition *with background and highlights* ($3.86 \pm 0.15$) as in the condition *with background, no highlights* ($3.89 \pm 0.10$), both of which are much higher ($p=0.03$) than the condition *without background* ($3.55 \pm 0.12$).

In summary, we find that highlighting relevant parts of the background does not enable users to critically assess incorrect model predictions and

leaves users prone to overconfidence in their inaccurate judgments.

**Subjectivity in User Responses** One concern with our analysis is the potential effect of participants' subjectivity in their agreement and confidence responses. To control for this, we fit a linear mixed-effect model, treating participants as a random factor for each of the aforementioned effects. In each mixed-effect model, we consider the measure of interest (that is, agreement, accuracy, or confidence) as the dependent variable and the treatment (presence of background, highlight, etc.) as the independent variable, along with the participant ID as a random effect. We find that even after controlling for the random effect of the variation among participants, the effect of background or highlights on users' reliance and confidence remains as discussed.

## 5 Related Work

Despite the promise of AI systems to assist humans in decision-making tasks (Kamar et al., 2012; Kamar, 2016), recent studies show that human-AI collaboration exhibits a persistent failure mode: humans tend to over-rely on AI assistance rather than using their own insights and reasoning when performing tasks (Chen et al., 2023). This over-reliance on AI systems can lead to the acceptance of incorrect suggestions (Bansal et al., 2021; Bussone et al., 2015; Jacobs et al., 2021; Lai and Tan,

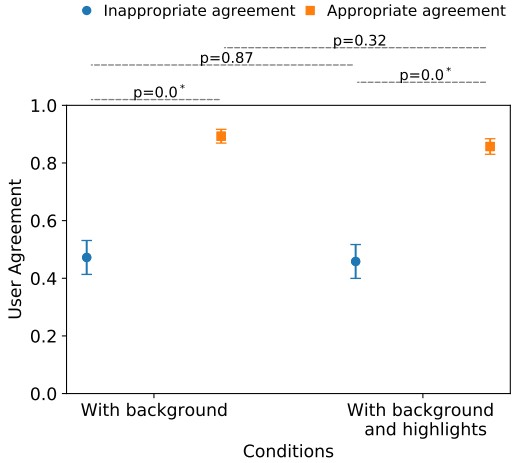 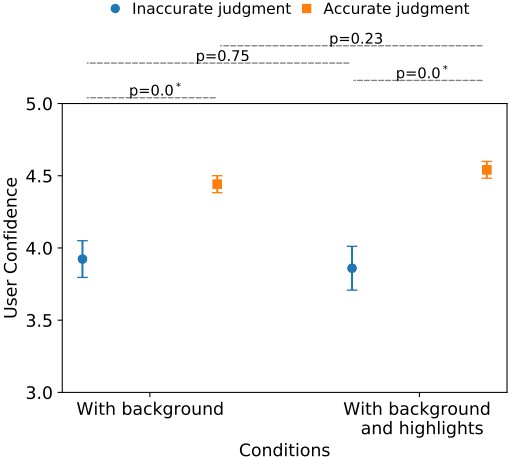

Figure 4: User agreement rate (mean and standard error) with correct vs. incorrect model predictions (left) and user confidence (mean and standard error) in their own accurate vs. inaccurate judgments (right) in the conditions *with background, no highlight* and the conditions *with background, supporting facts highlighted*. The rate of agreement is higher for correct model predictions (appropriate agreement) than for incorrect model predictions (inappropriate agreement), both with and without highlights. However, highlights do not help alleviate users' over-reliance on incorrect model prediction or overconfidence in their inaccurate judgments (*RQ4*).[6]

2019), a phenomenon that is of particular concern in high-stake domains where the continued use of AI systems is prevalent (Bussone et al., 2015; Jacobs et al., 2021; Lai and Tan, 2019).

To address the issue of over-reliance, recent efforts have focused on the development of explainable AI methods, with the aim of enhancing users' comprehension of AI decisions and, in turn, promoting more appropriate reliance (Bussone et al., 2015). However, an extensive body of human-centered evaluations of current explainability methods has revealed that explanations can inadvertently increase trust in AI models, exacerbating over-reliance (Bansal et al., 2021; Buçinca et al., 2020; Papenmeier et al., 2019; Schemmer et al., 2022; Jacobs et al., 2021; Zhang et al., 2020; Buçinca et al., 2021), particularly among non-expert users (Gaube et al., 2022; Schaffer et al., 2019).

Recent research in the field of explainable NLP has advocated for explanations that prioritize users' decision-making (Fok and Weld, 2023). Fok and Weld (2023) extensively review the current literature on the role of explanations in supporting human-AI decision-making, consolidating their findings into a compelling argument that explanations are most valuable when they empower users with the essential information required to critically evaluate the correctness of model predictions. This growing body of work underscores the pivotal function of explanations in facilitating users to scruti-

nize the accuracy of model predictions.

Nonetheless, the concept of explainability is frequently framed as providing "information necessary for the underlying model to make the correct prediction" which may not inherently align with "information sufficient for users to assess the model prediction." This misalignment is particularly relevant in contemporary question-answering (QA) systems, powered by large language models pre-trained on extensive datasets. These models can extrapolate knowledge beyond their immediate input context, drawing on implicit factual or commonsense knowledge (Petroni et al., 2019; Jiang et al., 2020) and employing shortcuts to provide accurate answers (Min et al., 2019; Chen and Durrett, 2019; Trivedi et al., 2020). While this ability enhances the performance of QA systems, it introduces complexities in the realm of explainability and user interaction with AI decisions.

Our work extends the landscape of explainability and human-AI decision-making by moving beyond model explanations and exploring the implications of providing users with relevant background information as a way to enhance verifiability and foster appropriate reliance on AI, in the context of question-answering.

## 6   Discussion and Conclusion

Large general-purpose language models, such as the GPT family of models (Brown et al., 2020;

OpenAI, 2023), LaMDA (Thoppilan et al., 2022), PaLM (Chowdhery et al., 2022; Anil et al., 2023), and others, have propagated into information-seeking workflows of a general audience. A vast host of existing and ongoing work in NLP examines the deficiencies of these language models, ranging from hallucinated generations (Bang et al., 2023; Ji et al., 2023), lack of transparency (Weld and Bansal, 2019; Lipton, 2018), reasoning gaps (Bang et al., 2023; Press et al., 2022), and more. However, close examination of the other piece of the puzzle—the user—is fairly sparse. Literature in Explainable NLP considers how NLP systems, and their underlying reasoning process, can be made available to the user to allow for better, more informed use of these systems and their predictions (Ribeiro et al., 2016; Wang and Yin, 2021). Extending this view of explainability, we considered the question of aiding users in their decision-making process, not by explaining the model prediction but by providing them with relevant pieces of information to assess the prediction. As argued by the contemporary work by Fok and Weld (2023), explanations in the form of the model's internal reasoning are rarely useful for supporting human decision-making (Bansal et al., 2021; Buçinca et al., 2021; Zhang et al., 2020). Instead, explanations should aim to help the users assess the model prediction. Our study takes a step in this direction by analyzing users' trust and reliance on QA models by providing information external to the model's input to enable verification of model predictions.

We conducted our study in a multi-hop question-answering setup. We designed our study such that some part of the information required to perform the reasoning is missing. We studied how users rely on model predictions in the absence of such information and how the presence of background affects their reliance and confidence in the QA systems. This background information, although not necessarily faithful to the model's reasoning process, may still be helpful to users in assessing the correctness of model predictions (Lipton, 2018). Such background explanations can take many shapes—implicit information encoded in the pre-trained models used for reasoning or external information required to fill in knowledge gaps.

Our study revealed that users' reliance on model predictions is fairly high (over $80\%$), much higher than the accuracy of the underlying QA system ($76\%$), even without sufficient information to as-

sess the correctness of the predictions (that is, without the relevant background). This indicates unwarranted trust (Jacovi et al., 2021) in the model to some extent (RQ1). We found that although background information did not affect reliance on correct model predictions, it significantly reduced over-reliance on incorrect model predictions (RQ2). This indicates that even though users trust the model without sufficient information, they are able to catch model errors better when provided with the relevant information. We further find that users are able to calibrate their reliance on model prediction even when the background is sometimes, but not always, perfect (RQ3).

On the flip side, our study revealed that the addition of background also increased users' confidence in their judgments of the model predictions, both those that are accurate and, unfortunately, those that are inaccurate. This is especially concerning as even if users are better at the task when the background information is available to them, uncalibrated confidence might be detrimental in critical decision-making tasks. Lastly, we found that surfacing relevant pieces to ease users' perusal of the background is not sufficient to alleviate overconfidence stemming from the background, even when the background and highlights pointed out the inaccuracies in users' judgments (RQ4).

Our work highlights the utility and pitfalls of adding more information with model predictions. We studied user interaction with such insufficient background information in a Wizard-of-Oz experimental setting. More work is needed to study how such information can be gathered or generated automatically and made available to the user. Research in NLP frequently adopts the reverse strategy—building the systems first, followed by testing if the systems are useful to the end-users. Our work aims to put humans front and center, studying their interactions with model predictions to inform the gaps in AI-assisted decision-making.

Our work highlights the issue of overconfidence stemming from more information, even when that information provides users with evidence to make accurate judgments. This indicates that explanations alone are not sufficient to garner appropriate trust and reliance. More efforts are needed to educate the lay audience to inspect explanations critically and diligently.

## Limitations

In this work, we conduct a Wizard-of-Oz experiment to study users' reliance and confidence in QA predictions in the lack of sufficient information and the effects of adding background information on their interaction with QA predictions. Wizard-of-Oz experimental setting allows us to control the amount of information and possible noise in the information provided to the user. But, this comes at a cost—constraining our study to simpler settings and model. User interaction with NLP systems is increasingly shifting to long-form question answering. We do not expect our findings to directly translate to this new paradigm, but we hope that our setting and findings will guide future exploration of these problems in the paradigm of long-form, open-ended question answering.

Our study relies on careful construction of background using a multi-hop question answering dataset (HotpotQA) that aligns with our assumptions: the question should require multiple pieces of evidence to reason the answer and the provided supporting documents should be sufficient to answer the question. The HotpotQA dataset had been gathered using human effort and is likely also prone to human errors, in the same vein as the humans interacting with the QA system in our study. This also makes our findings prone to the eccentricities or noise in the original dataset. We ensure insofar as possible that our study design and resulting examples align with our research questions. We posit that the constraints imposed in our study, such as using the same set of examples across conditions and ensuring consistency of answers with and without background, will possibly balance out any potential issues in the underlying data.

## Ethics Statement

The study conducted in this work was approved by the University of Maryland Institutional Review Board. We recorded minimal demographic information in our survey (age and gender). The demographic information was self-identified by the participants, was completely optional, and was not used to determine participation in the study. We only used the demographic information to gauge the participation distribution and did not perform any estimation of participants' demographic properties. We obtained informed consent from participants at the start of the study (see Appendix E for details on the consent form).

## Acknowledgements

We would like to thank Chen Zhao, Simone Stumpf, the anonymous reviewers, and members of the CLIP lab at UMD for their constructive feedback. This work was funded in part by U.S. Army Grant No. W911NF2120076.

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

## A  Data and Preprocessing

As described in Section 3, we use BERT large model fine-tuned on the SQuAD dataset as our QA model. We use the HotpotQA dataset (Yang et al., 2018), which has $90447$ and $7405$ questions in the training and validation split, respectively, to select examples for our study. We select examples from the training split of the dataset, allowing a larger candidate pool of examples that meet our criteria. This should not be a concern as the fine-tuned SQuAD model used in our work is not trained on the HotpotQA dataset. We remove question with *yes* and *no* type answers, leaving us with a candidate pool size of $84966$. Further, we only select examples for which the model answer is same with and without background. This leaves us with a candidate pool of $66772$ examples.

Lastly, we sample examples where the model confidence is between the range $[0.55, 0.65]$. The intuition behind this is that if the model confidence is too low, then the model can possibly refrain from answering. On the other hand, if the model confidence is fairly high, we might not need a human-in-the-loop to assess the model prediction. This ensures that there is a scope of meaningful collaboration between the human and the model, with their strengths complimenting each other (Heer, 2019; Zhang et al., 2020; Feng and Boyd-Graber, 2019). The threshold is chosen empirically such that the accuracy of the model on the selected range matches the aggregate accuracy of the model. This leaves us with a final candidate pool of 6212 examples.

## B  Subjective Assessment

In the post-task survey, we collect the following self-reported subjective measures to study users' overall perception of the model (Hoffman et al., 2018). For each measure, users are shown the corresponding statement and asked to rate each on a five-point Likert scale from strongly disagree to strongly agree:

○ *Usefulness of highlights:* "The highlights were useful. I feel that highlights helped in determining whether the model predictions were correct."

| Question | True answer / Predicted answer | Input context | Germane background | Ungermane background |
|---|---|---|---|---|
| Which song from a self-titled album was featured in a 1965 film, starring Jon Voight and written by Waldo Salt? | A Famous Myth / A Famous Myth | **The Groop (US band).** The Groop were a harmony-based psychedelic pop and soul vocal quartet from the USA, active at the end of the 1960s and releasing one self-titled album. Their song 'A Famous Myth' was included on the contemporary Midnight Cowboy film soundtrack. | **Midnight Cowboy.** Midnight Cowboy is a 1969 American drama film based on the 1965 novel of the same name by James Leo Herlihy. The film was written by Waldo Salt, directed by John Schlesinger, and stars Jon Voight alongside Dustin Hoffman. Notable smaller roles are filled by Sylvia Miles, John McGiver, Brenda Vaccaro, Bob Balaban, Jennifer Salt and Barnard Hughes. | **The Last of His Tribe.** The Last of His Tribe is a 1992 film starring Jon Voight as the anthropologist Alfred L. Kroeber who befriended Ishi, played by Graham Greene. Harry Hook directed the film. Ishi was thought to be the last of the Yahi people. |
| The Special Division was upheld in a case that decided which act was Constitutional? | Independent Counsel Act / Independent Counsel Act | **Morrison v. Olson.** Morrison v. Olson, 487 U.S. 654 (1988), is a United States federal court case in which the Supreme Court of the United States decided that the Independent Counsel Act was constitutional. | **Special Division.** The Special Division is a division of the United States Court of Appeals for the District of Columbia Circuit. 28 U.S.C. § 49 (1982 ed., Supp. V) (Title VI of the Ethics in Government Act). It consists of three circuit court judges or justices appointed by the Chief Justice of the United States. The judges are appointed for 2-year terms, with any vacancy being filled only for the remainder of the 2-year period. Its constitutionality was upheld in Morrison v. Olson. | **The Virginia Sterilization Act of 1924.** The Virginia Sterilization Act of 1924 greatly influenced the development of eugenics in the twentieth century. The act was based on model legislation written by Harry H. Laughlin and challenged by the Supreme Court decision of Buck v. Bell. The Supreme Court upheld the law; consequently, proving that it was constitutional and making it model law for sterilization laws in other states. |
| Love stars which Taiwanese actress and model? | Shu Qi / Shu Qi | **Shu Qi.** Lin Li-hui (born 16 April 1976), better known by her stage name Shu Qi, is a Taiwanese actress and model. She has also been credited as Hsu Chi and Shu Kei (Cantonese pronunciation of "Shu Qi"). She is among the highest paid actresses in China. | **Love (2012 film).** Love is a 2012 Taiwanese-Chinese romance film directed and co-written by Doze Niu. It stars Zhao Wei, Shu Qi, Mark Chao, Ethan Juan, Eddie Peng, Amber Kuo, Ivy Chen and Doze Niu. "Love" premiered in the Panorama section of the 62nd Berlin International Film Festival. The film features an ensemble cast, with the stories revealed to be interwoven as the plot progresses. | **Lorene Ren.** Lorene Ren (born 22 November 1988), previously known as Kirsten Ren, is a Taiwanese actress, model and singer. Her surname is sometimes spelled as Jen. She is the younger sister of Taiwanese girl group S.H.E member Selina Jen. Ren graduated from National Taiwan Normal University, with a bachelor's degree in home economics, human development and family studies. |
| For which platform did Prakhar Gupta write that closed following Rajaji's death? | Swarajya / South Asian Voices | **Prakhar Gupta.** Prakhar Gupta is an Indian journalist and a foreign affairs analyst. He has written on issues such as maritime security in the Indian Ocean Region, Territorial disputes in the South China Sea, Nuclear non-proliferation, and India-China-Pakistan relations. He has written for The Diplomat, Youth Ki Awaaz, The Frustrated Indian, Swarajya and South Asian Voices, an online platform for strategic analysis on South Asia hosted by The Stimson Center. | **Swarajya (magazine).** Swarajya is a monthly print magazine and online daily. It was a weekly magazine founded in 1956 by Khasa Subba Rao with the patronage of C. Rajagopalachari, one of the founders of the Swatantra Party, and a regular contributor to the magazine in the form of his "Dear Reader" column. Minoo Masani, R.Venkatraman, R.K. Laxman are some notable personalities who contributed to the magazine. After Rajaji's death in 1972, the magazine began to decline and eventually closed in 1980. | **C. Rajagopalachari.** Chakravarti Rajagopalachari BR (10 December 1878 – 25 December 1972), popularly known as Rajaji or C.R., also known as Mootharignar Rajaji (Rajaji, the Scholar Emeritus), was an Indian statesman, writer, lawyer, and independence activist. Rajagopalachari was the last Governor-General of India, as India became a republic in 1950. He was also the only Indian-born Governor-General, as all previous holders of the post were British nationals. |
| What actor, who first appeared in the Blackadder episode "The Foretelling" also played the role of Captain Darling in the same series? | Tim McInnerny / Rowan Atkinson | **The Foretelling.** "The Foretelling" is the first episode of the BBC sitcom 'The Black Adder'. the first series of the long-running comedy programme 'Blackadder'. It marks Rowan Atkinson's debut as the character Edmund Blackadder, and is the first appearance of the recurring characters Baldrick (Tony Robinson) and Percy (Tim McInnerny). The comedy actor Peter Cook guest stars as King Richard III. | **Tim McInnerny.** Tim McInnerny (born 18 September 1956) is an English actor. He is known for his many roles on television and stage. Early in his career he featured as Lord Percy Percy and Captain Darling in the "Blackadder" series. | **Rowan Atkinson.** Rowan Sebastian Atkinson (born 6 January 1955) is an English actor, comedian and writer. He played the title roles in the sitcoms Blackadder (1983–1989) and Mr. Bean (1990–1995), and in the film series Johnny English (2003–2018). |

Table 1: Examples: question, the true and predicted answers, along with the input context and the supporting (Germane) background with the supporting facts highlighted, and a distractor (Ungermane) background.

○ *Usefulness of background:* "The background information was useful. I feel that background helped in determining whether the model predictions were correct."

○ *Confidence in the model:* "I am confident in the model, including the predictions, highlights, and background."

○ *Self-confidence:* "I am confident in my decisions."

○ *Satisfaction with the model:* "I would like to use the model for decision-making."

**Does users' perception of the model change with and without background, or with highlights?** Figure 5 shows the user rating for the subjective measures across different conditions. We find that the differences in user ratings with and without background, both with and without highlights, are non-significant. We find that adding background increases users' confidence in their own judgment (from $3.74 \pm 0.16$ to $4.10 \pm 0.13$), consistent with users' aggregate confidence over individual examples (Figure 2 (right)). Despite the decrease in the rate of over-reliance on incorrect model predictions with background explanations (RQ1), users rate the satisfaction with the model marginally lower in the condition *with background* ($2.91 \pm 0.15$) than the condition *without background* ($3.13 \pm 0.16$). This might be due to the additional cognitive load re-

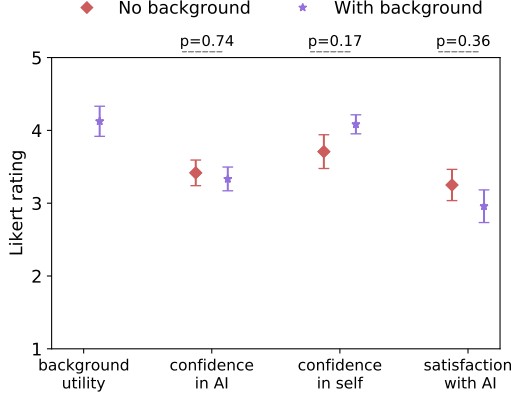 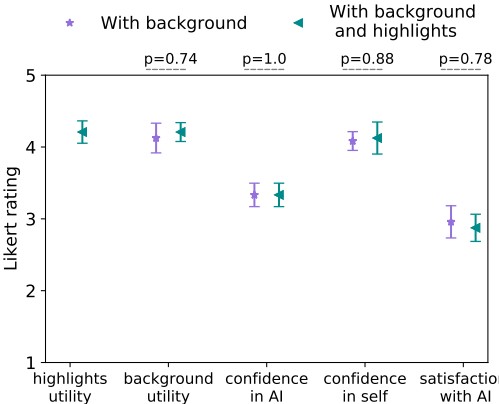

Figure 5: Users' subjective rating of the system for the usefulness of highlights, background, their confidence in the model, self-confidence, and satisfaction with the model. Users rate self-confidence marginally higher in the condition *with background* than the condition *without* (left). However, users rate their satisfaction with the model in the condition *with background* slightly lower than *without*. Users' satisfaction rating is slightly lower even after introducing highlights with the background (right), with a slightly higher rating of background utility in the condition *with highlights* than *without*. However, there are no other discernible differences in ratings in the background condition *with or without highlight*.

quired to consume the background information.

The differences in users' ratings are much less prominent for adding highlights along with background: users assign similar ratings for confidence in the model predictions and confidence in self-judgments with or without highlights. This parallels our findings on task performance, where we see negligible improvements in adding highlights. Surprisingly, users rate the satisfaction with the model in the condition *with highlight* marginally lower ($2.94 \pm 0.15$) than the condition *without highlights* ($3.10 \pm 0.16$).

## C Comparing Germane Background in the With Background vs. the Mixed Background Conditions

In RQ3 (Figure 3), we discuss the difference in users' reliance and confidence when they are shown germane background vs. ungermane background in the *mixed background condition*. We further compare whether and how users' reliance on model predictions and confidence in their judgments differ in the *with background condition*, when all examples they are shown have germane background, and *the mixed background* condition, when only half the examples shown have germane background. This allows us to see if users' interaction with relevant and useful background is affected by the presence of noise in the background information that is available to them.

As seen in Figure 6 (left), we find that users ex-

hibit comparable rate of appropriate agreement and inappropriate agreement in both the cases. This indicates that the presence of ungermane background examples in the *mixed background condition* does not affect users ability to calibrate their reliance on model predictions when they are shown useful background information.

Further, comparing the confidence in accurate and inaccurate judgments in the two cases (Figure 6 (right)), we find that users' exhibit a slightly lower confidence in their judgments even on predictions that include germane background in the *mixed background condition* than the *with background condition*. Unfortunately, this effect is not in the desirable direction. The users' confidence in their accurate judgments is significantly lower ($p = 0.0$) in the *mixed background condition* ($4.12 \pm 0.09$) than the *with background condition* ($4.44 \pm 0.06$). But, the drop in confidence in inaccurate judgments (from $3.92 \pm 0.13$ to $3.70 \pm 0.22$) across the two conditions is not significant ($p = 0.36$).

## D Comparing Ungermane Background with the No Background Condition

Extending the previous analysis, we further compare the difference in users' reliance and confidence on model predictions when they are shown ungermane background, which is topical, but not useful (in the *mixed background condition*), vs. when they are shown no background at all (in the *no background condition*). In both of these cases,

users likely lack sufficient information to assess the correctness of model prediction. This allows us to see whether our findings in RQ2 with respect to decrease in over-reliance and increase in over-confidence on the addition of background holds regardless of the utility of the background.

As seen in Figure 7(left), we observe that users' have a marginally higher rate of inappropriate agreement, and a significantly lower rate of appropriate agreement in the cases when the background is ungermane, as compared to the *no background condition*. This indicates that users do much worse with ungermane background than no background at all. In fact, users have a significantly lower accuracy when they are shown ungermane background in the *mixed background condition* $(0.65 \pm 0.04)$ as compared to when they are shown no background at all $(0.78 \pm 0.03)$.

Further, comparing confidence in the two cases (Figure 7(right)), we find that users also exhibit a significantly lower confidence in both their accurate (from $4.44 \pm 0.06$ to $3.67 \pm 0.15$; $p = 0.0$) and inaccurate (from $3.92 \pm 0.13$ to $3.16 \pm 0.17$; $p = 0.0$) judgments with ungermane background than no background at all.

## E   Study Interface

**Briefing and Consent**   At the start of the study, we brief the participants with the following details about the purpose of the study and obtain their consent to participate.

◦ *About the Study: The purpose of this study is to investigate ways to improve human and machine collaboration in decision-making tasks using explainable automated reasoning.*

◦ *Data and Confidentiality: The study collects minimal demographic information, such as age and gender. You can opt-out of answering demographic questions. Crowdworker ID is the only potentially identifying information, which will be immediately removed from the data after participants are paid. All data will be anonymized prior to public release and responses cannot be linked to individuals. Any potential loss of confidentiality will be minimized by storing data in a secured password protected database.*

◦ *Right to Withdraw And Questions: Participation in the study is completely voluntary. If you decide to participate in this research, you may stop participating at any time. If you have questions, concerns, or complaints, or if you need to report*

*an injury related to the research, please contact the principal investigator.*

**Instructions**   After starting the study, the participants are given the following instructions about the task:

◦ *In this survey, you will be presented with 10 questions and the respective answer predicted by an AI system based on the shown supporting document.*

◦ *You will also be shown background document (on the right) with additional information on the question and/or the supporting document.*

◦ *For each question, you will be asked to mark whether or not you agree with the AI predicted answer.*

◦ *The AI system is 70% accurate, that is, it predicts the correct answer in 70% cases.*

Next, participants are given a walk-through of the task. Figure 8a shows the example tutorial screen. After the tutorial, participants are shown 10 tasks, one by one. Figure 8b shows an example task screen for a participant in the *with background condition*. The study includes two attention check questions interspersed between the 10 tasks:

◦ *Did you agree with the AI in the previous answer? (Yes/No)*

◦ *What was your confidence rating in the last question? (Very Uncertain/ Uncertain/ Neutral/ Certain/ Very Certain)*

Lastly, participants are asked to fill in the post-task survey, shown in Figure 8c, followed by optional feedback and demographic information.

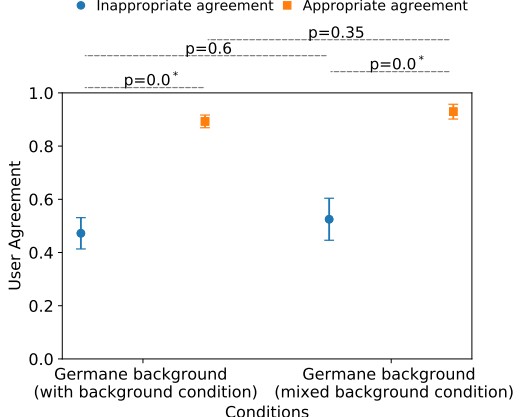
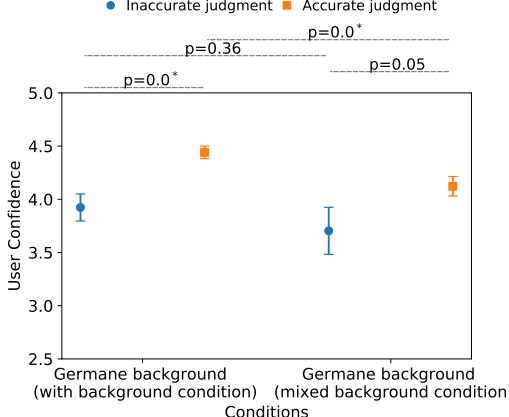

Figure 6: User agreement rate (mean and standard error) with correct vs. incorrect model predictions (left) and user confidence (mean and standard error) in their own accurate vs. inaccurate judgments (right) when they are shown the germane background in the *with background condition* (i.e., all examples have germane background) vs. the *mixed background condition* (i.e., only half the examples have germane background). We see no significant difference in the germane background in either background conditions, except users' confidence in their accurate judgments is lower in the mixed background condition, even on the examples that include the germane background.[6]

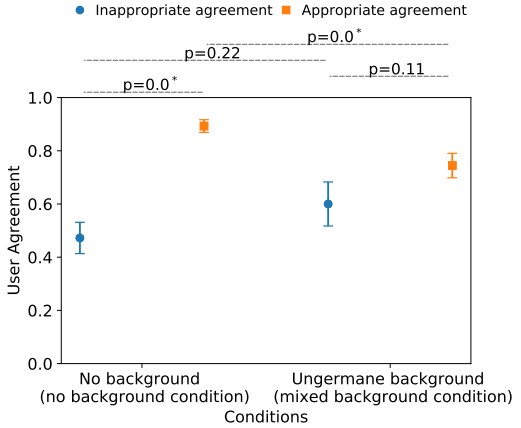
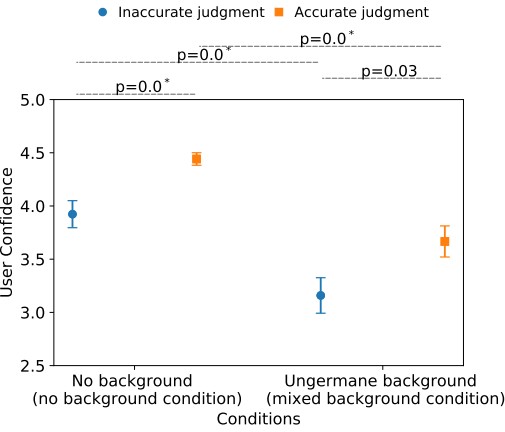

Figure 7: User agreement rate (mean and standard error) with correct vs. incorrect model predictions (left) and user confidence (mean and standard error) in their own accurate vs. inaccurate judgments (right) without background (i.e., in the *No background condition*) vs. with ungermane background (i.e., in the *mixed background condition*). We see that users' have a marginally higher rate of inappropriate agreement, and a significantly lower rate of appropriate agreement in the cases when the background is ungermane, as compared to the *no background condition*. Further, users also have a significantly lower confidence in both their accurate and inaccurate judgment with ungermane background than no background at all.[6]

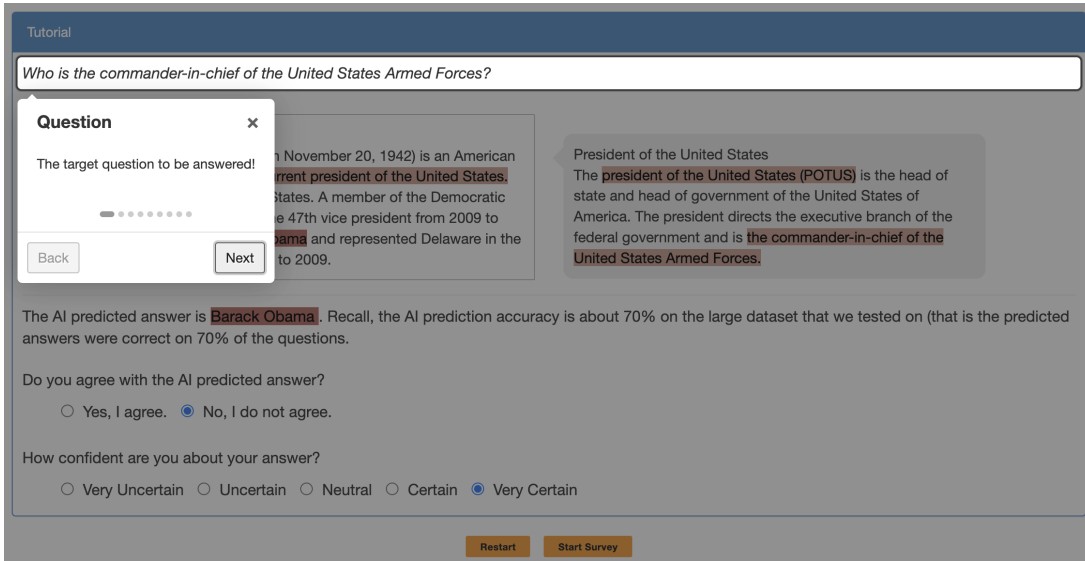

(a) Tutorial Screen. Participants are shown a tutorial example first. They are step-by-step taken through the different information that is shown in each example, such as the question, the supporting context, the background, and the model predicted answer.

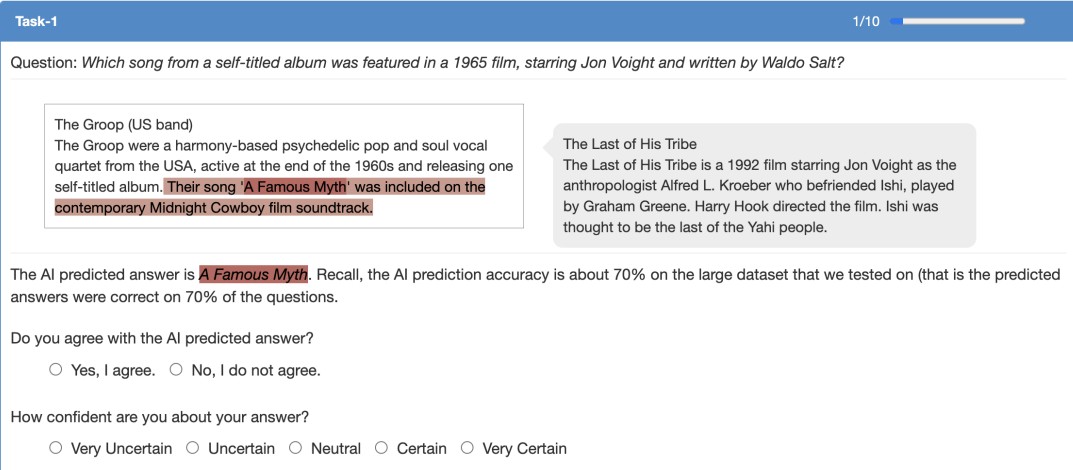

(b) Task Screen. One example task with the question, the supporting context (left) and the background (right). Participants are shown the model predicted answer and asked to specify their agreement/disagreement with the model's answer and indicate their level of confidence.

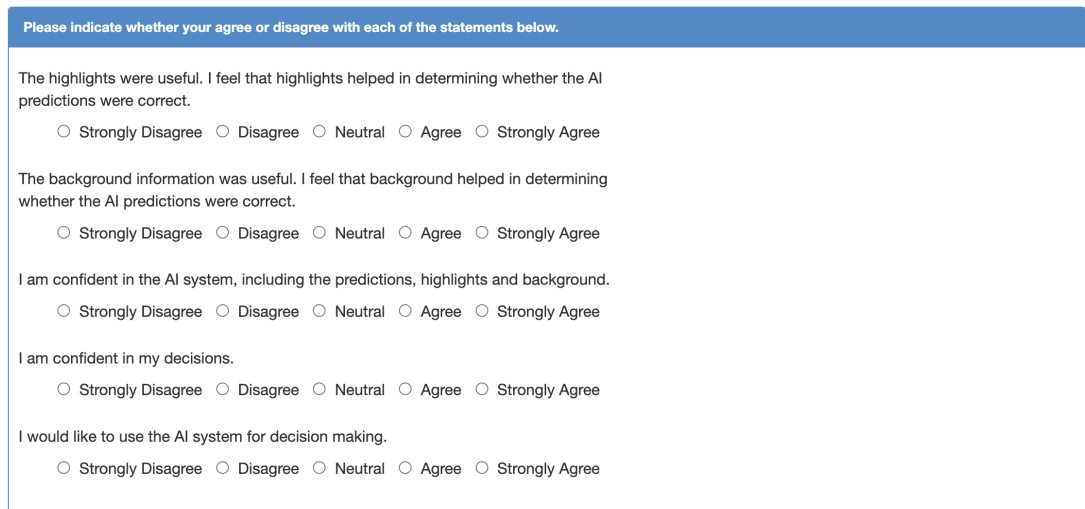

(c) Post-task Survey Screen. Participants are asked to provide aggregate ratings for highlights, background, confidence in self, confidence in the model, and satisfaction with the model.

Figure 8: Study Interface Screens