# OpenReview forum: "What Else Do I Need to Know? The Effect of Background Information on Users’ Reliance on QA Systems"
_EMNLP/2023/Conference — EMNLP 2023 Main_

### Official Review · Reviewer_7seo · 2023-08-03

**Soundness:** 3

**Excitement:**

3: Ambivalent: It has merits (e.g., it reports state-of-the-art results, the idea is nice), but there are key weaknesses (e.g., it describes incremental work), and it can significantly benefit from another round of revision. However, I won't object to accepting it if my co-reviewers champion it.

**Paper Topic And Main Contributions:**

The paper studies the importance of additional knowledge of users when assessing the correctness of QA systems. The authors analyze the agreement and confidence of users, with respect to automatic systems, depending on the additional information provided to the users.
The main contribution is to evaluate how additional documents affect the confidence and agreement of users with respect to answers automatically extracted.

**Reasons To Accept:**

- Study of the effect of additional documents on the agreement and confidence of users
- Study of the impact of including germane and ungermane background

**Reasons To Reject:**

- The work is, in some way, limited to results of only one model with a 76% of accuracy. I think the study would benefit from including another model with lower accuracy. Thus, we could compare the results of users' behavior with different models' accuracy.

- The objective of the study is clear, but the benefits/implications of your study in other works are unclear. I find some of them at the end of the conclusions but such a work should include more in the Introduction and along the paper. For example, the paper would benefit from some recommendations after the results of each experiment. Otherwise, it is a little bit hard to understand the impact of this work.

- Although this work includes many references inserted across the document, I miss a related work section, besides what is included in the Introduction and Discussion Section.

**Reproducibility:**

4: Could mostly reproduce the results, but there may be some variation because of sample variance or minor variations in their interpretation of the protocol or method.

**Reviewer Confidence:**

5: Positive that my evaluation is correct. I read the paper very carefully and I am very familiar with related work.

**Typos Grammar Style And Presentation Improvements:**

- Section 2 contains more than the research questions. I think a more proper name would be "Experiment design" or something similar
- you should introduce the list of Items from line 365 to line 372.

---

> ### Author Rebuttal · Authors · 2023-08-28
>
> Thank you for your comments and suggestions on our work. We really appreciate your feedback both on the study and the writing.
>
> Re the results being limited to results of only one model with a 76% accuracy: Our work aims to study the effect of the amount of information presented to the users on their reliance and confidence in the system, leading us to focus more on this axis of variation. We choose a model with 76% accuracy for this study as, in this case, the model accuracy is consistent both  with and without background explanation, which is one key desideratum in our study. Overall, as the users are not aware of the instances that the model is correct/incorrect on, the model accuracy only guides their aggregate behavior, at best. Since the accuracy is maintained across the conditions with and without background, our findings with respect to the differences in users’ reliance on correct/incorrect predictions across conditions would hold.
>
> Re the implications of our  findings: The key takeaway of our work is that aiding users in assessing model predictions is indeed useful in helping them better weed out incorrect predictions, but it comes at the cost of increased overconfidence in inaccurate judgments. This is relevant for current NLP systems, which, despite good performance, are known to be erroneous. The expectation is often that providing users with the necessary information to assess model predictions would be sufficient in combating model deficiencies. But as our work highlights, this is clearly not sufficient. Thus, more thought needs to be put into the best ways of aiding users in analyzing model predictions and evidence/explanations. We will incorporate the reviewer’s feedback and bring forth these points in the introduction and the discussion of the results.
>
> Re the absence of a separate related work section: we will add this in the final version of the paper.

---

### Official Review · Reviewer_Dbbf · 2023-08-05

**Soundness:** 4

**Excitement:**

4: Strong: This paper deepens the understanding of some phenomenon or lowers the barriers to an existing research direction.

**Paper Topic And Main Contributions:**

The authors conduct a wizard-of-oz experiment that investigates how sufficient or insufficient background information influences users’ reliance on QA systems and self-confidence. They restrict the focus to two-hop questions in HotPotQA, where two passages are used to answer the question, in order to control for with or without full background knowledge. They find that users tend to agree with model prediction even when they are wrong (over-reliance), and that having access to background information could mitigate that. However, access to background information also leads to over-confident in human judgements. Highlighting relevant sentences in the background information makes little difference.

**Questions For The Authors:**

Question A: How do you distinguish if the user is making the judgment depending on the model prediction or the background? (When there is background passage) It is likely that the user misread the background and make incorrect judgments. Is it possible to conduct extra experiments to find out about this?
Question B: Different participants are recruited in different background settings. How do you ensure that the agreement / confidence differences between different settings come from design, not from the difference in participants themselves?


**Reasons To Accept:**

1. The experiments are solid: The design of various background settings are appropriate for the research questions. The experimental results are shown with mean, variance and p-value. The study comes with a tutorial and simple filtering mechanism.
2. Could be useful for future work: Few works have discussed how humans make decisions (judgements) about whether to trust a model prediction. The insights gained from this work could be used for better QA model design. This work could also inspire future work in this  particular direction.
3. Clear writing: The experimental design is carefully laid out and easy to follow. The research questions are listed explicitly and explained clearly. All technical terms are well-explained.


**Reasons To Reject:**

1.  Generalizability: The results in this paper may not generalize to different settings. The experiments are only conducted in one dataset, and only two-hop questions are considered. It is difficult to determine if the findings would hold across different domains, answer types (e.g. long-form QA vs. short factoid QA), and reasoning types.

**Reproducibility:**

3: Could reproduce the results with some difficulty. The settings of parameters are underspecified or subjectively determined; the training/evaluation data are not widely available.

**Reviewer Confidence:**

3: Pretty sure, but there's a chance I missed something. Although I have a good feel for this area in general, I did not carefully check the paper's details, e.g., the math, experimental design, or novelty.

---

> ### Author Rebuttal · Authors · 2023-08-28
>
> We thank the reviewer for their comments and suggestions.
>
> Re generalization to different settings: Given that this is the first study that explores the use of background information to enable verification of model prediction, we considered the trade-off between in-the-wild model generated explanations and wizard-of-oz explanations and argued that the latter allows a more controlled experimental setting. With wizard-of-oz experiments, we control the sufficiency of information available to the participants, which isn’t possible in model-generated explanations.
>
> Secondly, there isn’t yet a clear and definitive way to evaluate long-form answers or explanations, which might then conflate model and evaluation errors and human errors. Indeed, more work is needed to extend this kind of evaluation and studies to other settings. Extending this to more naturalistic settings is part of an ongoing future work. Our work here aims to motivate this line of investigation and tools, common in HCI literature, to the NLP community.
>
> (Question A) How do you distinguish if the user is making the judgment depending on the model prediction or the background: We do not directly control whether the user makes the judgment based on the model prediction or background. It would have been entirely possible that users entirely ignore the background, in which case we would have observed no differences in users’ reliance or confidence between the with and without background conditions. It is also possible that users entirely ignore the model prediction in the with background condition. Regardless, a decrease in the over-reliance on incorrect predictions still indicates the utility of background in helping users make better decisions, as  there are no significant differences in the time taken to complete the task with and without background. To further validate this, we can check whether the users are able to do the task as efficiently and accurately without model predictions altogether. Due to IRB constraints, we can not run this study within the rebuttal time frame, but we will add a discussion in the final paper.
>
> (Question B) Different participants are recruited in different background settings. How do you ensure that the agreement / confidence differences between different settings come from design, not from the difference in participants themselves: We recruit a large set of participants (100) who are assigned randomly to the different conditions, so the expectation is that the significant differences observed across conditions come from the design.  To further ascertain this, we run a new analysis, where we fit a mixed effect model that considers the assigned participants as the random effect. We find that even after controlling for the random effect of the variation among participants, the effect of background on users’ reliance and confidence remains as discussed in the paper: background reduces over-reliance on incorrect AI predictions, but also increases overconfidence in incorrect judgments. Further, we calculate the intra class correlation statistics, which measure the similarity of the agreement/confidence ratings within a participant (random effect). The intra class correlation (ICC) lies between 0 and 1, where 1 indicates a perfect relationship within a cluster. We find the intra class correlation to be 0.04 and 0.07, respectively, for the agreement and confidence. In our case, this reflects that there is large heterogeneity in the responses of a single participant, indicating that individual participant’s judgments vary based on the question and evidence. We will add this discussion in the final version of the paper.

---

### Official Review · Reviewer_bvZJ · 2023-08-07

**Typos Grammar Style And Presentation Improvements:** For clarity, maybe consider making th…
**Soundness:** 5

**Excitement:**

4: Strong: This paper deepens the understanding of some phenomenon or lowers the barriers to an existing research direction.

**Paper Topic And Main Contributions:**

This work presents a study on user interactions with QA systems, specifically investigating the relationship between user confidence in a QA system's output and the amount of evidence shown the the user. The authors are particularly interested in multi-hop settings where the answer cannot be completely answered from the answer's context paragraph alone and require additional information in a background paragraph. They experiment with showing the users (1) the context alone, (2) the context and correct background, (3) the context and mixed correct/incorrect backgrounds, and (4) the correct background with supporting facts highlighted.

In each of these settings, the authors ask users (1) whether they agree with the QA system's output and (2) how confident they are in that judgment. The authors find that users, across all settings, tend to agree with correct outputs more often that incorrect ones. They also find that adding correct background information increases users' ability to identify correct/incorrect predictions; however, their confidence in all judgments (correct and erroneous) also increases. In the mixed correct/incorrect background settings, the authors observe a similar trend where confidence increases for all judgments when the context is correct/relevant. In their experiments with highlighting relevant information, the authors find that this does not improve overconfidence from users.

**Reasons To Accept:**

This paper investigates a well-motivated, under-explored topic in the intersection of HCI and QA.

The user studies are well designed, and significance tests are conducted through each experiment. The authors also are thorough in reporting their crowd working setup for their user study.

This papers findings identify a core issue in existing QA systems, one that may motivate further work into designing effective explanations for real-world users.

**Reasons To Reject:**

While this paper identifies issues in QA system explanations, they do not experiment with any methods that effectively alleviate this issue (outside of highlighting which did not work). They suggest possible small fixes (such as not providing evidence when its irrelevant); however these are still unsatisfactory long-term solutions and not implemented in this work.

It difficult to interpret the different options for confidence in the user study. I'd be curious to see data on how the users interpreted each confidence option, either in their post-task survey or analyzing how average confidence judgments differed user-to-user. In particular, I'm curious whether users associated "neutral" confidence more with 50% confidence in their judgment or with 75% confidence (which seems to be the intended interpretation from the authors).

Biggest concern is the limited breadth of the study, both with respect to the types of models/explanations and types of questions. The authors do not explore results with LLMs and model generated explanations, which are far more common than systems that produce multiple retrieved evidence passages. The authors also only explain multihop questions which may not represent a realistic distribution of information-seeking questions from users.

**Reproducibility:**

4: Could mostly reproduce the results, but there may be some variation because of sample variance or minor variations in their interpretation of the protocol or method.

**Reviewer Confidence:**

4: Quite sure. I tried to check the important points carefully. It's unlikely, though conceivable, that I missed something that should affect my ratings.

---

> ### Author Rebuttal · Authors · 2023-08-28
>
> We sincerely thank the reviewer for their comments and for highlighting the key strengths of our work. We really appreciate your time in reading through this work and providing feedback. Here are some thoughts on the points mentioned in your review.
>
> Re absence of methods that alleviate issues in QA system explanations (besides highlighting which does not work): We agree with the reviewer that our paper mainly highlights how providing background information required to assess QA predictions helps (decreasing over-reliance) and hurts (increasing overconfidence) users’ interaction with QA systems and does not provide a solution to alleviate this issue. Our work aims to encourage caution in treating explanations as a sure-fire fix to human-AI decision making and call for thinking more deeply about the shortcomings of solutions that are commonly sought after in literature (Fok and Weld, 2023) and motivating further research in potential fixes that better calibrate human reliance and confidence.
>
> Re variability in the interpretation of confidence scores: Yes, there might be some variations in the interpretation of the confidence scores between participants. We did not find anything of note regarding the interpretation of confidence score in the end user survey.
>
> To assess the effect of participants’ subjectivity in interpreting the confidence score, we run a new analysis, where we fit a mixed effect model that considers the assigned participants as the random effect. We find that even after controlling for the random effect of the variation among participants, the effect of background on users’ reliance and confidence remains as discussed in the paper: background reduces over-reliance on incorrect AI predictions, but also increases overconfidence in incorrect judgments.  Further, we also calculate the intra class correlation statistics, which measures the similarity of the agreement/confidence ratings within a participant (random effect). The intra class correlation (ICC) lies between 0 and 1, where 1 indicates a perfect relationship within a participant. We find the intra class correlations to be 0.04 and 0.07, respectively, for the agreement and confidence. In our case, this reflects that there is large heterogeneity in the responses of a single participant, indicating that individual participant’s judgments vary based on the question and evidence. We will add this discussion in the final version of the paper.
>
> Re model-generated explanations and long-form QA: We considered the trade-off between in-the-wild model generated explanations and wizard-of-oz explanations and argued that the latter allows a more controlled experimental setting for a first study. With wizard-of-oz experiments, we control the sufficiency of information available to the participants, which isn’t possible in model-generated explanations.
>
> Secondly, there isn’t yet a clear and definitive way to evaluate long-form answers or explanations, which might then conflate model and evaluation errors and human errors. Indeed, more work is needed to extend this kind of evaluation and studies to other settings. Extending this to more naturalistic settings is part of an ongoing future work. Our work here aims to motivate this line of investigation and tools, common in HCI literature, to the NLP community.

---

### Meta-Review · Area_Chair_9w5A · 2023-09-18

**Recommendation:** 4

**Metareview:**

The paper uses a wizard-of-oz experiment using two-hop questions from HotpotQA, to study how people's reliance on QA model predictions change when they have access to (in-)sufficient context knowledge.

All reviewers agree that the study is interesting and well-motivated, and that the experiments are well-designed and well-executed.

Main concerns involve:
1. Generalizability: Whether the findings generalize to other domains or QA tasks
2. Implications: The study flags a phenonmenon but did not provide a solution

The authors sufficiently addressed these concerns in their rebuttals. I recommend accepting the papers to the Main track, and recommend the authors to revise the paper as they proposed in their rebuttals: add related work discussions, reflect on potential biases in human studies (e.g., whether they actually used the given information and whether their confidences are well-calibrated), and discuss the implications on futuer work.

---

### Decision · Program_Chairs · 2023-10-07

**Decision:**

Accept-Main

**Comment:**

The paper uses a wizard-of-oz experiment using two-hop questions from HotpotQA, to study how people's reliance on QA model predictions change when they have access to (in-)sufficient context knowledge.

All reviewers agree that the study is interesting and well-motivated, and that the experiments are well-designed and well-executed.

Main concerns involve:
1. Generalizability: Whether the findings generalize to other domains or QA tasks
2. Implications: The study flags a phenonmenon but did not provide a solution

The authors sufficiently addressed these concerns in their rebuttals. I recommend accepting the papers to the Main track, and recommend the authors to revise the paper as they proposed in their rebuttals: add related work discussions, reflect on potential biases in human studies (e.g., whether they actually used the given information and whether their confidences are well-calibrated), and discuss the implications on futuer work.